# Dynamics and Endocytosis of Flot1 in *Arabidopsis* Require CPI1 Function

**DOI:** 10.3390/ijms21051552

**Published:** 2020-02-25

**Authors:** Yangyang Cao, Qizouhong He, Zengxing Qi, Yan Zhang, Liang Lu, Jingyuan Xue, Junling Li, Ruili Li

**Affiliations:** 1Beijing Advanced Innovation Center for Tree Breeding by Molecular Design, Beijing Forestry University, Beijing 100083, China; 2College of Biological Sciences and Technology, Beijing Forestry University, Beijing 100083, China

**Keywords:** Flot1, sterols, VA-TIRFM, endocytosis, plant immunity

## Abstract

Membrane microdomains are nano-scale domains (10–200 nm) enriched in sterols and sphingolipids. They have many important biological functions, including vesicle transport, endocytosis, and pathogen invasion. A previous study reported that the membrane microdomain-associated protein Flotillin1 (Flot1) was involved in plant development in *Arabidopsis thaliana*; however, whether sterols affect the plant immunity conveyed by Flot1 is unknown. Here, we showed that the root length in sterol-deficient *cyclopropylsterol isomerase 1* (*cpi1-1*) mutants expressing Flot1 was significantly shorter than in control seedlings. The cotyledon epidermal cells in *cpi1-1* mutants expressing Flot1 were smaller than in controls. Moreover, variable-angle total internal reflection fluorescence microscopy (VA-TIRFM) and single-particle tracking (SPT) analysis demonstrated that the long-distance Flot1-GFP movement was decreased significantly in *cpi1-1* mutants compared with the control seedlings. Meanwhile, the value of the diffusion coefficient Ĝ was dramatically decreased in *cpi1-1* mutants after flagelin22 (flg22) treatment compared with the control seedlings, indicating that sterols affect the lateral mobility of Flot1-GFP within the plasma membrane. Importantly, using confocal microscopy, we determined that the endocytosis of Flot1-GFP was decreased in *cpi1-1* mutants, which was confirmed by fluorescence cross spectroscopy (FCS) analysis. Hence, these results demonstrate that sterol composition plays a critical role in the plant defense responses of Flot1.

## 1. Introduction

Membrane microdomains perform a vast array of functions, including signal transduction, membrane trafficking, organization of the actin cytoskeleton, and responses to pathogen attacks [1,2,3,4]. Proteins containing the stomation/prohibitin/flotillin/hypersensitive-induced reaction (HIR) (SPFH) domain are found in divergent species, ranging from prokaryotes to eukaryotes, and they are enriched in membrane microdomains [5,6,7]. In the past few years, numerous studies have demonstrated that these SPFH domain-containing proteins are involved in biological processes, such as the cellular localization of proteins, the regulation of ion channels, the formation of membrane microdomains, vesicle transport, and interaction between the membrane and the cytoskeleton [5,7,8,9,10]. In *Arabidopsis thaliana*, the overexpression of pepper CaHIR1 enhanced resistance to *Pseudomonas syringae* pv. *tomato* (*Pst*) and *Hyaloperonospora parasitica* [11]. More importantly, Choi et al. (2011) reported that CaHIR1 plays an important role in plant disease and immunity as a positive regulator of cell death [12]. Further analysis suggested that AtHIR proteins participate in RPS2-mediated effector-triggered immunity (ETI) to *Pto DC3000 AvrRpt2* by forming a complex with RPS2 protein [13]. Recently, wheat TaHIR1 and TaHIR3 were reported to have important functions in stripe rust fungus infection and abiotic stresses [14]. Remorins are typical microdomain proteins which also play roles in plant defense. Recent data strongly suggested that remorin oligomers could control infection in *Medicago truncatula* and the release of rhizobia into the host cytoplasm [15]. AvrRPM1 is a well-known effector in plants that can enhance the pathogenicity of pathogens and inhibit pathogen-associated molecular patterns (PAMPs)-triggered immunity (PTI). The expression level of AtREM1.2 was significantly increased when *AvrRPM1* was overexpressed, indicating that remorin participated in ETI in *Arabidopsis* [16]. RIN4 is a plasma membrane protein involved in PTI and ETI in plants. A previous study showed that AtREM1.2 directly interacted with the RIN4 protein. These data illustrated that AtREM1.2 was related to plant disease resistance [16]. Flotillin has been suggested to play a critical role in endocytosis and plant immune signaling. Li et al. (2012) reported Flot1 was involved in a clathrin-independent endocytic pathway and is required for seedling development [17]. Recent evidence suggested that Flot2 and Flot4 are involved in the infection of nitrogen-fixing bacteria in *Medicago truncatula* [18]. In addition, Yu et al. (2017) demonstrated that the dynamics and aggregation of Flot1-GFP in plasma membrane can contribute to flg22-induced endocytosis and degradation of Flot1 in *Arabidopsis* [19].

Membrane microdomains are highly dynamic domains enriched in sterols and sphingolipids on the plasma membrane [20]. Methyl-β-cyclodextrin (MβCD) can deplete sterols from the plasma membrane, and MβCD-induced membrane depletion has been described as a characteristic of sterol-dependent proteins [21]. Previous studies have shown that MβCD removes sterols from membrane microdomains in a concentration-dependent manner [22]. Several studies have shown that the MβCD-responsive proteins include a great number of cell wall-related proteins, fasciclin-like arabinogalactan proteins, and glycosyl-hydrolase family proteins, most of which have been shown to be glycosylphosphatidylinositol-anchored [23]. Interestingly, sterol depletion by MβCD treatment attenuated dynamics, phosphorylation, dimerization, and internalization, of the plant blue light receptor phototropin 1 (phot1), suggesting that membrane microdomains serve as signaling platforms for phot1 [24]. Furthermore, treatment with MβCD induced dramatic changes in the partitioning of GFP-PIP2;1 in the plasma membrane, and well-dispersed patterns of diffraction-limited spots disappeared [25]. After treatment with mβCD, the distribution density of AMT1;3-EGFP on the plasma membrane increased significantly, indicating that the membrane microdomains on the plasma membrane were disrupted, inhibiting the dissociation of membrane proteins from the plasma membrane [26]. In addition, fenpropimorph (Fen), a sterol synthesis inhibitor, and DL-threo-1-phenyl-2-palmitoylamino-3-morpholino-1-propanol (PPMP), a sphingolipid biosynthesis inhibitor, can deplete sterols from the membrane microdomains. Some particles accumulated into small clusters and the fluorescence intensity of PIP2;1 increased after treatment with either Fen or PPMP [25]. Moreover, treatment with all of the inhibitors, including Fen and PPMP, activated long-distance AtHIR1 movement and caused a significant increase in the diffusion coefficient [27]. In addition, the endocytosis of BRI1-GFP was also significantly inhibited after treatment with PPMP [28].

In addition, it has been reported, that studies have showed that defective mutants of enzymes in the sterol biosynthetic pathway, such as *cvp-1*, *fackel* (*fk*), *hydra1* (*hyd1*) and *smt1* mutants, disturb cell division plane orientation [29,30], epidermal morphology [31] and the polar localization of PIN protein [32,33,34]. It is worth mentioning that *Arabidopsis* CYCLOPROPYLSTEROL ISOMERASE 1 (CPI1) is encoded by a single-copy gene and displays cycloeucalenol-obtusifoliol isomerase activity in vitro [35]. A trap line with Ds transposons was found in the third intron of the *CPI1* gene, known as *cpi1-1* [36]. The sterol profile of *cpi1-1* seedling roots and the whole plants showed a substantial conversion [36]. In addition, different cyclopropylsterols accounted for 99% of the total sterol content in homozygous *cpi1-1* plants, and the wild-type sterols, including sitosterol, sosterol, rapeseed sterol, and isosterol, were almost completely reduced [36]. Men et al. (2008) found that *Arabidopsis cpi1-1* mutants were defective in polarization of PIN2 localization, and endocytosis was decreased [36]. As a membrane microdomain-associated protein, Flot1 was demonstrated to be involved in the endocytosis and required for seedling development in *Arabidopsis* [17]. However, it’s still inconclusive whether sterols have an effect on the biological function of Flot1 protein. In particular, the dynamics and endocytosis of Flot1 in *cpi1-1* remains to be further elucidated.

In the present study, we aimed to demonstrate that the effects of CPI1-1 on the function of Flot1 protein. Moreover, using VA-TIRFM and SPT analysis, we compared the dynamics of Flot1-GFP in control and *cpi1-1* mutants. In addition, we performed an integrative analysis of Flot1 endocytosis in *cpi1-1* mutants when stimulated by the plant immunity stimulant flg22. These findings provide new insights into the link between sterols and Flot1 dynamics in plant defense.

## 2. Results

### 2.1. The Subcellular Localization of Flot1-GFP Fusion Protein in cpi1-1 Mutants

To investigate the behavior of Flot1 in *Arabidopsis thaliana cpi1-1* mutants, we transformed *cpi1-1* mutants with *Flot1*-*GFP* under the control of the endogenous *Flot1* promoter. Successful transformation was confirmed by PCR using gene-specific primers (Figure 1A, Appendix A). 

The root length of *cpi1-1* mutants expressing Flot1 was reduced in 84% when compared to the control seedlings (Figure 1B,C). To gain further insight into the basis for this retarded growth, we stained root cells with FM4-64 and observed the cells under laser scanning confocal microscope (LSCM) (Figure 1D). The results revealed that the roots of the *cpi1-1* mutants were significantly shorter than those of the control seedling.

We also detected Flot1-GFP fluorescence signal of cotyledon epidermal cells and found that Flot1-GFP was not uniform in control seedlings and *cpi1-1* mutants (Figure 2A). Interestingly, compared with the control plants, the Flot1-GFP in the *cpi1-1* mutants was not fully colocalized with the membrane marker FM4-64, indicating that Flot1 was distributed more than in the plasma membrane (Figure 2A). Furthermore, we observed the leaf epidermal characteristics of the *cpi1-1* mutants by scanning electron microscopy (SEM). As shown in Figure 2B, the leaf epidermis of the control seedlings consists of a two-dimensional array of interdigitated lobed pavement cells. However, the pavement cells in *cpi1-1* mutants have an altered shape; the pavement cells were more rounded, the protrusions and indentations were either small or absent, and the pavement cells were smaller (Figure 2B). The number of cotyledon epidermal cells in *cpi1-1* mutants was 153% of that in control seedlings (Figure 2C).

### 2.2. The Intensity of the Flot1-GFP Signal in cpi1-1 Mutants

We investigated the dynamics of individual Flot1 particles at the plasma membrane in cotyledon epidermal cells by VA-TIRFM. The Flot1-GFP particles were localized in the plasma membrane, where they occurred as separate spots with almost constant fluorescence and showed patchy structures (Figure 3A,B). Moreover, we performed a 10-sec recording in 0.1-sec intervals, and found that the individual particles represented stable structures with lateral and temporal dynamics (Figure 3a-1–b-2).

By SPT analysis, we found that the fluorescence intensity of Flot1-GFP spots in the *cpi1-1* mutants (25950.01 counts and 51498.24 counts) was lower than that in control seedlings (28328.46 counts and 58723.95 counts) (Figure 3C,D). Next, we measured the size of the single protein particles using ImageJ software. As shown in Figure 3E, the average size of Flot1-GFP spots was 3.69 × 3.69 ± 4.52 pixels in the *cpi1-1* mutants, significantly smaller than the protein spots in control seedlings, in which the average size was 3.88 × 3.88 ± 4.82 pixels, indicating that the depletion of sterols resulted in a reduction of Flot1-GFP clusters (Figure 3E). By analyzing the density of punctate structures of Flot1-GFP at the plasma membrane, we found that the density of Flot1-GFP in *cpi1-1* was 1.05 ± 0.07 N/µm^2^, whereas the density of Flot1-GFP in control seedlings was 0.71 ± 0.09 N/µm^2^, suggesting that the endocytosis of Flot1-GFP in *cpi1-1* mutants was significantly inhibited (Figure 3F).

### 2.3. The Dynamics of Flot1-GFP at the Plasma Membrane in cpi1-1 Mutants

To investigate whether sterols affect the diffusion of Flot1-GFP, we examined the motion ranges of Flot1-GFP in *cpi1-1* mutants. Under control conditions, the range of motion of Flot1-GFP showed a bimodal distribution: long-distance motion (77.06%, 0.54 ± 0.05 µm, the Gaussian peak value) and short-distance motion (22.94%, 0.13 ± 0.01 µm, the Gaussian peak value) (Figure 4A). In the *cpi1-1* mutants expressing Flot1-GFP, the percentage of Flot1-GFP exhibiting short-distance motion increased to 31.50% (0.13 ± 0.01 µm, the Gaussian peak value) and the percentage of Flot1-GFP exhibiting long-distance motion decreased to 68.50% (0.63 ± 0.06 µm, the Gaussian peak value), suggesting that the depletion of sterols in *cpi1-1* mutants significantly inhibited long-distance Flot1-GFP movement (Figure 4A,B). To further study the effects of sterols on the plant immunity of Flot1-GFP, we also compared the motion range and diffusion coefficients of Flot1-GFP after flg22 treatment. When the control seedlings were treated with flg22, long-distance and short-distance motion accounted for 82.56% (0.67 ± 0.04 µm, the Gaussian peak value) and 17.44% (0.15 ± 0.03 µm, the Gaussian peak value), respectively (Figure 4E). Notably, after flg22 treatment, the range of motion of Flot1-GFP was markedly different from the distribution in the control seedlings, with only 72.31% (0.53 ± 0.04 µm, the Gaussian peak value) of Flot1-GFP particles showing long-distance motion and 27.69% (0.14 ± 0.01 µm, the Gaussian peak value) of the particles showing short-distance motion in *cpi1-1* mutants (Figure 4E,F). These results indicated that sterol depletion inhibited the diffusion of Flot1-GFP into wider regions.

We further analyzed the distribution of diffusion coefficients of Flot1-GFP. Our results are illustrated in histograms and fitted using Gaussian functions, in which we defined the Gaussian peaks (Ĝ) as the characteristic values. In the control seedlings, the diffusion coefficients of Flot1-GFP were distributed into two populations with Ĝ values of 1.14 ± 0.35 × 10^−3^ µm^2^/s and 7.94 ± 0.72 × 10^−2^ µm^2^/s, respectively (Figure 4C). In *cpi1-1* mutants, the Ĝ value of diffusion coefficients was 1.32 ± 0.60 × 10^−3^ µm^2^/s and 1.45 ± 0.32 × 10^−1^ µm^2^/s, indicating that the *cpi1-1* mutation affected the diffusion coefficients of Flot1-GFP (Figure 4C,D). After flg22 treatment, the diffusion coefficient of Flot1-GFP in the control plants was 1.93 ± 0.45 × 10^−3^ µm^2^/s and 1.20 ± 0.19 × 10^−1^ µm^2^/s (Figure 4G). The distribution of diffusion coefficients of Flot1-GFP in *cpi1-1* plants after flg22 treatment also showed two populations, with Ĝ values of 1.65 ± 0.60 × 10^−3^ µm^2^/s and 7.30 ± 0.64 × 10^−2^ µm^2^/s, representing a significant decrease of diffusion coefficient in *cpi1-1* mutants compared with control seedlings after flg22 treatment (Figure 4G,H).

### 2.4. The Endocytosis of FLOT1-GFP in cpi1-1 Mutants

To gain insights into the impact of sterols on the endocytosis of Flot1-GFP in *Arabidopsis thaliana*, we stained the *cpi1-1* mutants and control seedlings with FM4-64. After treatment with FM4-64 for 30 min, the endocytosis of Flot1-GFP in the *cpi1-1* mutants (12.33%) was significantly decreased compared with the control seedlings (27.33%) (Figure 5A,B). 

To further probe whether the sterols affect the function of Flot1 in plant immunity, we analyzed the endocytosis of Flot1-GFP in *cpi1-1* mutants after pretreatment with flg22 for 60 min and incubation of the transgenic seedlings with FM4-64 for 30 min. We found that the colocalization of endosomes labeled with Flot1-GFP and FM4-64 was significantly decreased in the cotyledon epidermal cells in *cpi1-1* mutants (22.33%) compared with the control seedlings (34.00%) (Figure 5C,D). These results demonstrated that the sterols affect the endocytosis of Flot1 in the *Arabidopsis thaliana.*

We also applied FCS, which is an effective technique for monitoring of fluctuations in fluorescence intensity within the focal volume of the laser beam, to precisely measure the density of Flot1-GFP molecules under different conditions. Our FCS analyses showed that the mean density of Flot1-GFP molecules at the plasma membrane in the control seedlings was 10.0 ± 1.4 molecules μm^−2^. In *cpi1-1* mutants, the Flot1-GFP density was significantly increased to 17.9 ± 5.1 molecules μm^−2^ (79% increase with respect to control cells), indicating that more Flot1-GFP molecules accumulated in the plasma membrane in *cpi1-1* mutants (Figure 5F). In *cpi1-1* mutants, the mean density of Flot1-GFP molecules also showed a marked increase after flg22 stimulation compared with control plants (13.0 ± 2.4 molecules μm^−2^ vs. 7.1 ± 1.3 molecules μm^−2^, an 83% increase with respect to control cells), suggesting that the endocytosis of Flot1-GFP was significantly inhibited in *cpi1-1* mutants (Figure 5F).

## 3. Discussion

The membrane microdomain provides a functional platform for membrane trafficking, organization of the actin cytoskeleton, transduction of various signals, including defense responses [1,2,3,4]. In addition, various pathogens, including fungi, viruses, and eukaryotic parasites, can invade cells by the membrane microdomains [37,38]. By analyzing the composition of membrane microdomain in tobacco, Furt et al. showed that the phosphatidylinositol 4,5-bisphosphate [PI(4,5)P_2_] was located in membrane microdomain fractions [39]. A previous study reported that the level of PI(4,5)P_2_ in the extra-invasive hyphal membrane (EIHM) was modified by the *Colletotrichum higginsianum* (*Ch*), which could increase the exocytic membrane/protein supply of the EIHM for successful infection [40]. Flg22 is a peptide comprising 22 amino acids of the highly conserved elicitor-active epitope of flagellin, which can induce the immune response of plants [41]. A study found that, when elicited by flg22, flagellin sensing 2 (FLS2) and brassinosteroid-associated kinase 1 (BAK1) can form a complex, therefore activating downstream defense-signaling pathways [16,42,43,44]. Moreover, miR393 was found to regulate PTI induced by flg22 [45]. In addition, after flg22 treatment, proteins such as FLS2, Ca^2+^-dependent kinases, and syntaxin can relocalize to detergent-resistant membranes (DRMs) [46]. Upon stimulation with flg22, phosphorylation of RbohD was activated, and some regulatory factors, including small G proteins, are also localized to sterol-rich membrane domains [47]. Studies have shown that plant Flotillin plays a key role in the resistance against symbiotic bacterial infection [18]. In addition, based on the reported properties of Flot2 interactors, Flot2 complexes may be involved in plant–pathogen interactions, water transport and intracellular trafficking [48]. During an analysis of callose deposition, Yu et al. (2017) has reported that Flot1 amiRNAi mutants displays defects in immune responses induced by the flagellin-derived peptide flg22 [19]. In the present study, our results showed that the sterols affected the dynamics and endocytosis of Flot1-GFP in cotyledon epidermal cells when elicited by flg22. 

Plasma membranes are highly dynamic. Membrane proteins can diffuse within the plasma membrane and switch between multiple modes in equilibrium [49]. The dynamics of membrane proteins are thought to play vital roles in protein interactions, protein polymer formation, and signal transduction [50]. It’s a challenge to analyze the dynamics and activation of protein molecules within living cells. VA-TIRFM is a type of microscopy that confines the excitation to a depth of 100–400 nm in the specimen [51,52]. This is a powerful technique to clarify spatially the details of complex dynamics in living cell at the single molecule level [53]. Upon the salt treatment, both the diffusion coefficients and the percentage of restricted diffusion of GFP-PIP2;1 were increased [25]. The *Arabidopsis* Brassinosteroid Insensitive 1 (BRI1) plays a key role in brassinosteroid (BR) signaling as the BR receptor [54,55]. When treated with high concentrations of the BR analog 24-epibrassinolide (eBL), the motion range and diffusion coefficient of BRI1-GFP on the plasma membrane increased significantly, suggesting that BRI1-GFP displays highly heterogeneous mobility [28]. After treatment with high levels of blue light, the motion range and diffusion coefficient of phot1-GFP both increased, indicating that phot1-GFP particles moved faster and diffused into wider regions [24]. After JA treatment, the motion range and diffusion coefficient of AtRGS1-YFP were all increased significantly, suggesting that AtRGS1 exhibited increased lateral mobility within the plasma membrane [56]. Importantly, sterols are integral components of the cell membrane and affect the lateral mobility of plasma membrane proteins [57]. We spatially clarified the dynamics of Flot1-GFP in response to flg22 by VA-TIRFM. After flg22 treatment, the Flot1-GFP proteins in *cpi1-1* mutants moved over a much shorter distance when compared with the control seedlings, suggesting that sterol depletion can limit Flot1 particle diffusion into wide areas. After flg22 treatment, the diffusion coefficient of Flot1-GFP in *cpi1-1* mutants was decreased to 1.65 ± 0.60 × 10^−3^ µm^2^/s and 7.30 ± 0.64 × 10^−2^ µm^2^/s, revealing that flg22 treatment resulted in a significant decreased diffusion coefficient compared with the control seedlings. These results indicate that sterols affect the biological function of the protein by changing its dynamics.

Membrane proteins are important constituents of plasma membranes and regulate endocytic recycling to control the distribution of proteins at the cell surface [58]. Sterol-enriched domains may participate in pathogen-associated molecular pattern-induced signaling and the endocytosis of plasma membrane proteins in plants [59,60]. In *Arabidopsis thaliana*, sterols affect the endocytosis of the PIN2 auxin transporter, resulting in a lack of PIN2 polarity [36]. More importantly, sterols can regulate endocytic pathways during flg22-induced defense responses of FLS2 in *Arabidopsis* [61]. A recent study reported that flg22 can activate the endocytosis of Flot1-GFP in *Arabidopsis* [19]. FCS provides a powerful approach which can directly measure the densities of live cell membrane proteins in the native environment without affecting protein function [62]. In the present study, we observed a significant decrease in the number of endosomes labeled with GFP-Flot1 and FM4-64 in the cotyledon epidermal cells of *cpi1-1* mutants, suggesting that sterols also affect the endocytosis of Flot1-GFP. After treatment of transgenic plants with flg22, the number of endosomes labeled with Flot1-GFP and FM4-64 was also significantly decreased in *cpi1-1* mutants. In addition, FCS analysis revealed that the density of Flot1-GFP on the plasma membrane in *cpi1-1* mutants significantly increased upon flg22 treatment compared with control seedlings. These results are similar to a previous study, the endocytosis of FLS2 was inhibited in *smt1* mutants compared with control seedlings after flg22 treatment [61]. Therefore, we conclude that the sterols affect the function of proteins in disease resistance by affecting their endocytosis.

## 4. Materials and Methods

### 4.1. Plant Materials and Growth Conditions

The *cpi1-1*, wild-type, and Flot1-GFP transgenic *Arabidopsis thaliana* were all Landsberg *erecta* ecotype. Transgenic *Arabidopsis thaliana* seeds in this study were surface- sterilized with ethanol and H_2_O_2_ mixture (70% enthanol:30% H_2_O_2_ = 4:1 ) and sown onto half-strength Murashige and Skoog (MS) medium containing 1% (*w*/*v*) sucrose and 0.8% (*w*/*v*) agar. After stratification at 4 °C in the dark for 2 days, the seedlings were grown at 20–22 °C with a 16 h light/8 h dark cycle for 4 days. For root length measurements, the seeds were planted and grown on half-strength MS medium for 7 days.

### 4.2. Plasmid Construction and Plant Transformation

The Flot1-GFP plant expression vector was constructed as follows. The coding sequence for Flot1 was amplified from the *Arabidopsis* genome and subcloned as a BamHI–EcoRI fragment into a modified pCAMBIA1300 vector under the control of the native AtFlot1 promoter. Wild-type and *cpi1-1* plants were transformed with the 1300-Flot1-GFP expression vector by the *Agrobacterium tumefaciens*-mediated floral dip method [63], and transgenic seedlings were selected on half-strength MS solid medium (1% agar) containing kanamycin for the control seedlings and 70 mg/mL hygromycin for *cpi1-1* mutants. 

### 4.3. Scanning Electron Microscopy

The *Arabidopsis thaliana* plants were immersed in formalin–acetic acid–alcohol (FAA) fixative, vacuum-treated, and fixed in FAA fixative overnight. The samples were dehydrated with a series of gradient ethanol solutions (30%, 40%, 50%, 60%, 70%, 80%, and 90%) for 40 min, and the plants were dehydrated overnight with anhydrous ethanol. A dryer (EM CPD300, Leica, Germany) was used to treat dehydrated samples, and the dry samples were sputter-coated with gold (110 s). The pictures were taken with a scanning electron microscope (S-4800, HITACHI, Tokyo, Japan). The accuracy and efficiency of the region of interest (ROI) were improved with the following parameters: accelerating voltage, 10 kV; beam current, 10 µA.

### 4.4. Drug Treatments

The FM4-64 (5 mM in DMSO, working solution 5 µM) was purchased from Sigma-Aldrich (St. Louis, MO, USA). The flagellin peptide flg22 (10 mM in double distilled H_2_O, working solution 10 µM) was synthesized by GL Biochem (Shanghai, China).

### 4.5. Confocal Imaging

For confocal microscopy analysis, seeds were cultured under the conditions described previously for 4 days and then mounted in half-strength MS liquid medium. All of the samples were imaged using a FluoView 1200 inverted confocal microscope (Olympus, Japan) fitted with a 60× water-immersion objective (numerical aperture 1). Flot1-GFP and FM4-64 were all excited at 488 nm. Green signals were detected at 520–550 nm, while red signals were detected at 560–640 nm.

### 4.6. VA-TIRFM Live Imaging and Data Analysis

After flg22 treatment, the 4-day-old vertically grown transgenic *Arabidopsis* seedlings were transferred onto a glass slide with half-strength MS medium, covered with a coverslip, and observed under an objective-type variable-angle total internal reflection fluorescence microscope, which was based on an Olympus IX-71 microscope equipped with a total internal reflective fluorescence illuminator and a 100× oil-immersion objective (numerical aperture of 1.45, Olympus). GFP was excited by laser beams from a diode laser (Changchun New Industries Optoelectronics Technology, Changchun, China) at 473 nm. The fluorescence signals were collected by the objective, passed through filter (525/545), and detected by a back-illuminated electron multiplying charge-coupled device (EMCCD) camera (ANDOR iXon DV8897D-CS0-VP; Andor Technology, Belfast, UK). In all of the single-molecule imaging experiments, we set the gain of the EMCCD camera at 300. All of the images were acquired with 100-ms exposure time and analyzed with ImageJ software.

### 4.7. FCS

FCS was carried out on a Leica TCS SP5 FCS microscope equipped with a 488-nm argon laser, in-house coupled correlator, and an Avalanche photodiode. Flot1-GFP was excited at 488 nm with an argon laser through a 63× water immersion objective. The laser focus was placed at the plasma membrane of a cell in the measurement area. After acquiring images of cells in transmitted light mode, FCS was performed in the point-scanning mode. The diffusion of Flot1-GFP molecules into and out of the focal volume changed the local concentration of the fluorophore number, which led to spontaneous fluorophore intensity fluctuations. Flot1-GFP density was calculated according to a previously described protocol [25].

## 5. Conclusions

This study show that sterols is involved in Flot1-mediated plant defense response by inhibiting the internalization and coordinating dynamics of Flot1 proteins in the plasma membrane.

## Figures and Tables

**Figure 1 ijms-21-01552-f001:**
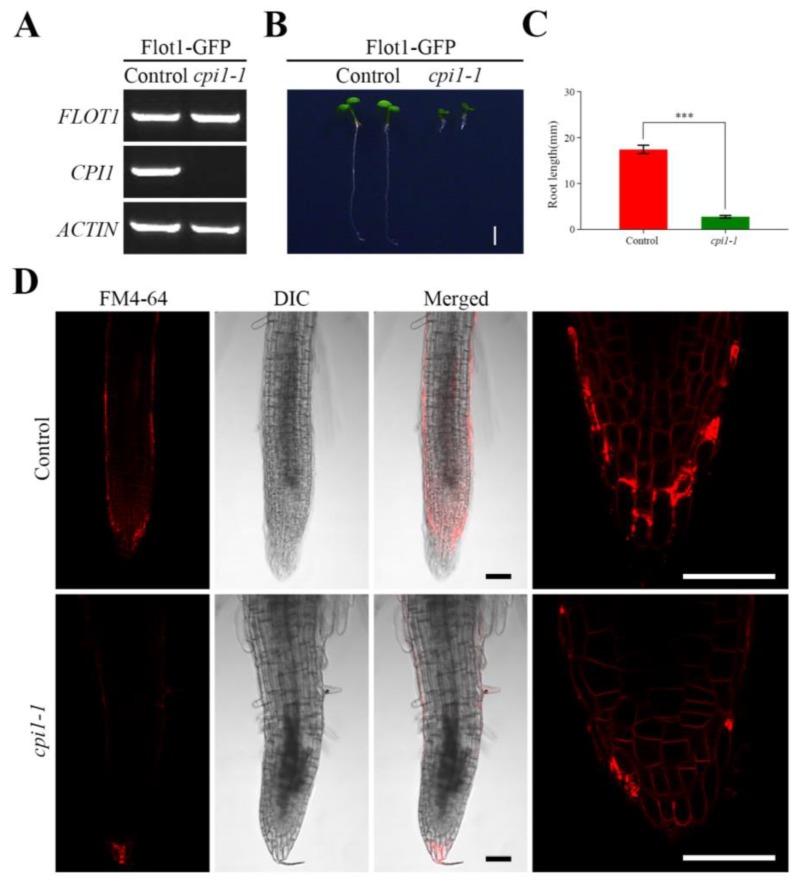
Characterization of the root in Flot1-GFP transgenic mutants. (**A**) Semiquantitative RT-PCR analysis of Flot1 expression in control seedlings and *cpi1-1* mutants. (**B**) A typical *cpi1-1* mutant plant and wild-type transgenic *Arabidopsis thaliana*. Scale bar, 0.25 cm. (**C**) The root length of Flot1-GFP/control and Flot1-GFP/*cpi1-1* mutants. Data shown are from 43 to 65 plants. Error bars represent the SD (*n* = 3). *** *p* < 0.001, Student’s *t*-test. (**D**) The shape of a typical root cell in *cpi1-1* mutants and wild-type transgenic *Arabidopsis thaliana*. Scale bars, 50 µm.

**Figure 2 ijms-21-01552-f002:**
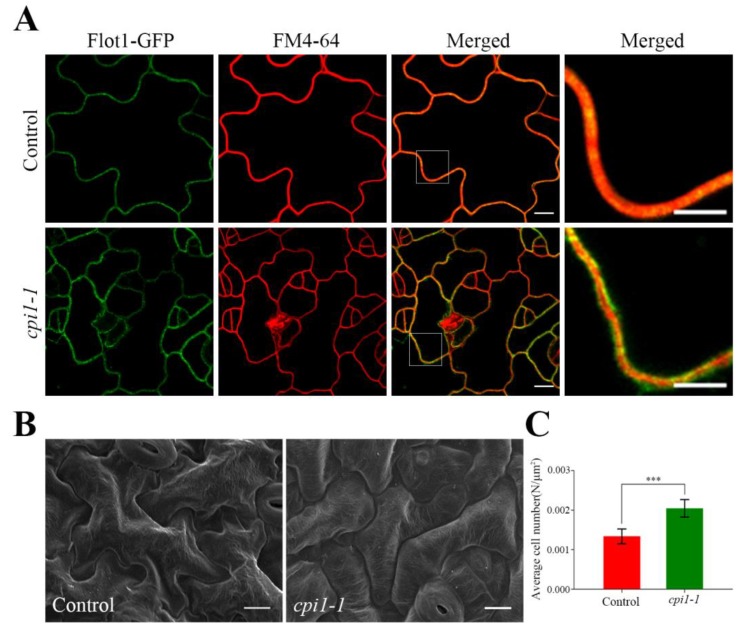
Characterization of the cotyledon epidermal cells in Flot1-GFP transgenic mutants. (**A**) The shape of a typical cotyledon epidermal cell in *cpi1-1* mutants and wild-type transgenic *Arabidopsis thaliana*. Scale bar, 10 µm. Localization of Flot1-GFP on the membrane of leaf epidermis cells. Scale bar, 5 µm. (**B**) Scanning electron microscopy images of *cpi1-1* mutants and control seedling cotyledon epidermal cells. Scale bar, 10 µm. (**C**) Average number of cotyledon epidermal cells of leaves. Data shown are from 24 to 30 plants. Error bars represent the SD (*n* = 3). *** *p* < 0.001, Student’s *t*-test.

**Figure 3 ijms-21-01552-f003:**
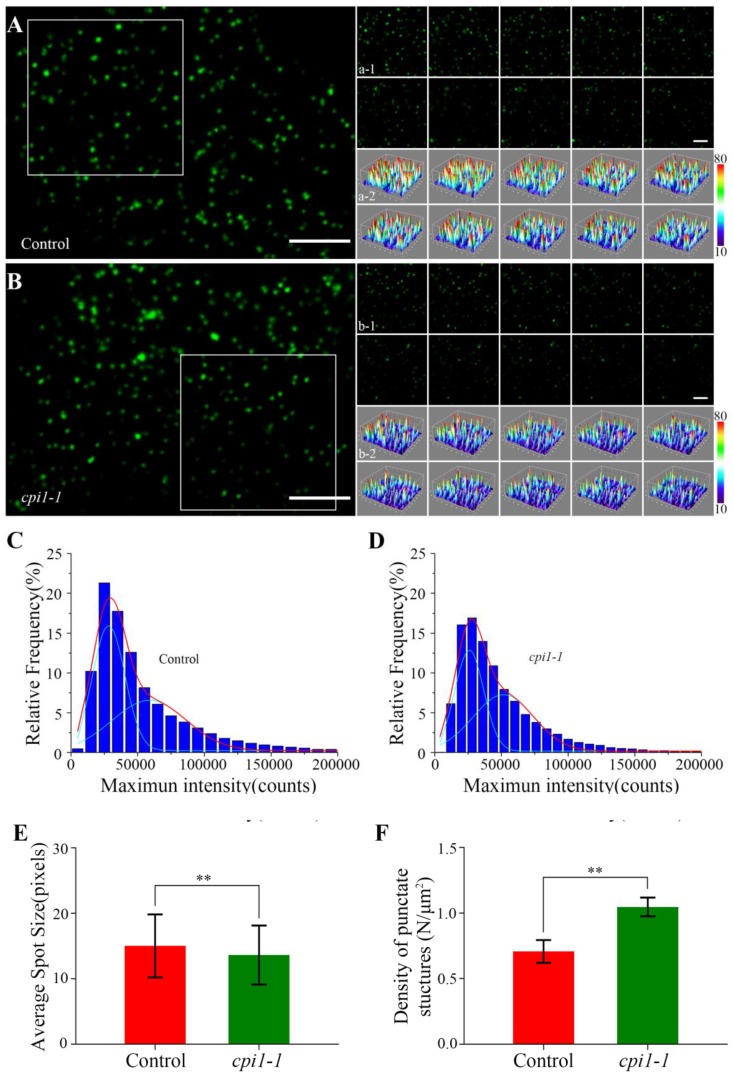
Fluorescence intensity on the membrane of Flot1-GFP expressing plants. (**A**) A typical single-particle image for Flot1-GFP at the plasma membrane in control seedlings. Scale bar, 2.5 µm. (a-1) Dynamic analysis and (a-2) three-dimensional luminance plots using a time series of the boxed area in (A). Scale bar, 2.5 µm. (**B**) A typical single-particle image for Flot1-GFP at the plasma membrane in *cpi1-1* mutants. (b-1) Dynamic analysis and (b-2) three-dimensional luminance plots using a time series of the boxed area in (B). (**C,D**) Maximum intensity of Flot1-GFP in (C) control (*n* = 5782 spots) and (D) *cpi1-1* mutant seedlings (*n* = 7196 spots). (**E**) Average spot size and (**F**) the density at the plasma membrane of Flot1-GFP fluorescent spots in *cpi1-1* mutants (*n* = 337 spots) and control seedlings (*n* = 296 spots). Error bars represent the SD (*n* = 4). ** *p* < 0.01, Student’s *t*-test.

**Figure 4 ijms-21-01552-f004:**
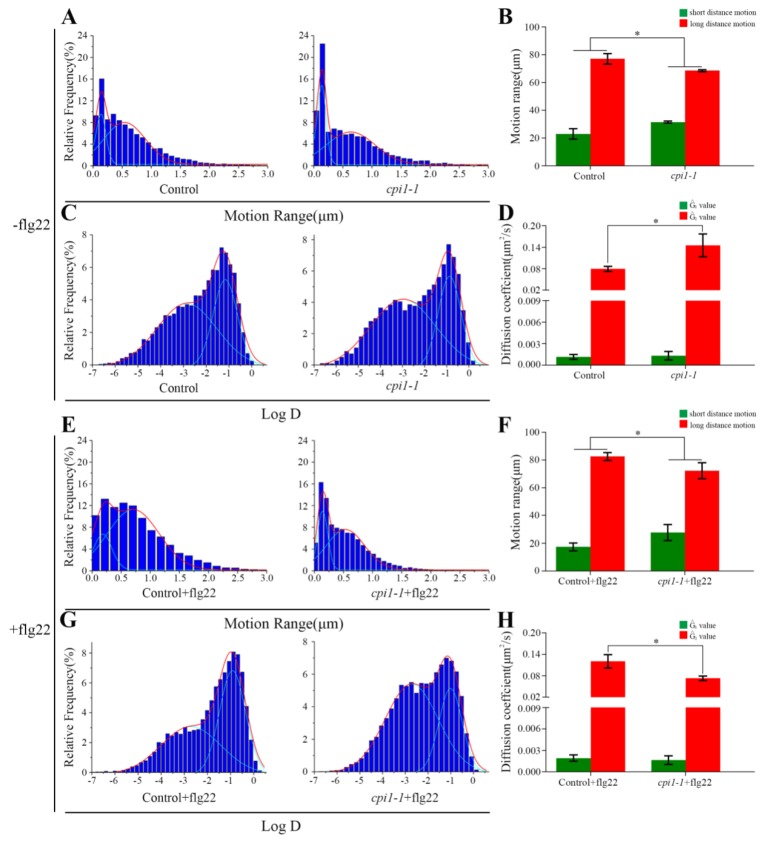
Effects of sterols on the dynamics of Flot1-GFP spots at the plasma membrane. (**A**) Distribution of Flot1-GFP motion range in control (*n* = 9987 spots) and in *cpi1-1* mutant (*n* = 5316 spots) seedlings. (**B**) Frequency of long-distance and short-distance motions for Flot1-GFP without flg22 treatment. Error bars represent the SD (*n* = 15–20). * *p* < 0.05, Student’s *t*-test. (**C**) Distribution of Flot1-GFP diffusion coefficients in control (*n* = 10389 spots) and in *cpi1-1* mutant (*n* = 5316 spots) seedlings. (**D**) Diffusion coefficients of Flot1-GFP without flg22 treatment. Error bars represent the SD (*n* = 15–20). * *p* < 0.05, Student’s *t*-test. (**E**) Distribution of Flot1-GFP motion range in control (*n* = 8896 spots) and in *cpi1-1* mutant (*n* = 7737 spots) seedlings after flg22 treatment (15 min). (**F**) Frequency of long-distance and short-distance motions of Flot1-GFP after flg22 treatment. Error bars represent the SD (*n* = 15–20). * *p* < 0.05, Student’s *t*-test. (**G**) Distribution of Flot1-GFP diffusion coefficients in control (*n* = 8896 spots) and in *cpi1-1* mutant (*n* = 7737 spots) seedlings after flg22 treatment (15 min). (**H**) Diffusion coefficients of Flot1-GFP after flg22 treatment. Error bars represent the SD (*n* = 15–20). * *p* < 0.05, Student’s *t*-test.

**Figure 5 ijms-21-01552-f005:**
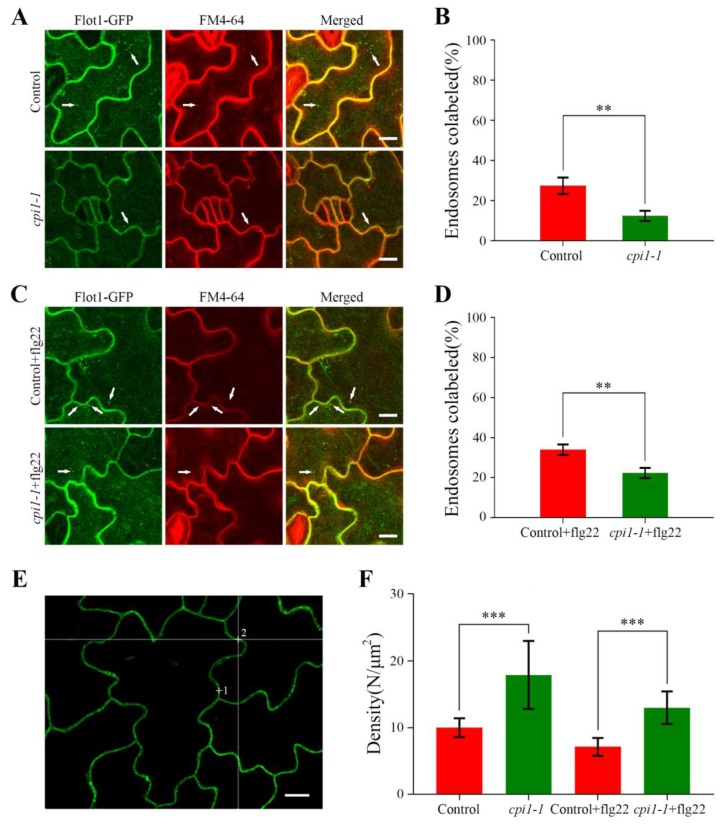
Confocal images showing endocytosis of Flot1-GFP in transgenic *Arabidopsis* lines. (**A**) Cotyledon epidermal cells were stained with FM4-64 and the endocytosis of Flot1-GFP and FM4-64 was observed in intracellular structures. White arrows indicate endosomes showing colocalization. Flot1-GFP is shown in green, FM4-64 in red, and the merged image indicating colocalization (yellow). Scale bar, 10 μm. (**B**) Quantification of endosomes colabeled in control and *cpi1-1* mutants cells. Data shown are from 27 to 42 plants. Error bars represent the SD (*n* = 5–8). ** *p* < 0.01, Student’s *t*-test. (**C**) Cotyledon epidermal cells were treated with flg22 (10 μM) for 60 min, followed by staining with FM4-64 for 30 min. The endocytosis of Flot1-GFP and FM4-64 was observed in intracellular structures. Flot1-GFP colocalized with FM4-64 was observed in the endosome (white arrow). Scale bar, 10 μm. (**D**) Quantification of endosomes colabeled in control and *cpi1-1* mutants cells after flg22 treatment. Data shown are from 28 to 38 plants. Error bars represent the SD (*n* = 5–8). ** *p* < 0.01, Student’s *t*-test. (E) Typical confocal image. The laser beam was focused on points 1 and 2 to monitor the fluorescence fluctuations. Scale bar, 10 µm. (**F**) The density of Flot1-GFP particles on the plasma membrane of cotyledon epidermal cells after different treatments. Data shown are from 32 to 47 plants. Error bars represent the SD (*n* = 4). *** *p* < 0.001, Student’s *t*-test.

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
