# Peer review of "Dynamics and Endocytosis of Flot1 in Arabidopsis Require CPI1 Function"

_ijms, 2020, doi:10.3390/ijms21051552_

Round 1
Reviewer 1 Report
Although this is an interesting work, in my opinion there are major drawbacks that should be addressed before publication. My main concerns regard the experimental design, discussion and conclusions, as well as the references, as described below:
(1) Gene names should be described in whenever a gene is cited for the first time, throughout the MS, e.g.:
Line 15: - “Flot1”
describe as you did for cpi1-1 Line 23: “flg22” – describe as for cip1-1(2) Check the scientific names, e.g. In line 15 “Arabidopsis thaliana” is not in italic
(3) Line 37: “biological process”: use plural

(4) Lines 60-81: I would reduce this paragraph (is the detailed description of the role of sterol inhibitors relevant?) 
and transfer it to the beginning of the introduction, when the membrane microdomains are introduced (lines 32-34), before presenting the SPFH 
proteins. I think it makes more sense to first explain what are microdomains and their constitution and only then refer the protein networks involved in the different biological processes and the research gaps.
(5) The link between cpi1-1 and smt1 (lines 89-97) is not clear; please refer this aspect or delete the information on smt1; it would be more important to explain the link between flot and cpi proteins (and thus the hypothesis of your research).
(6) Lines 98-107: rather than summarizing the results, the rationale of the research should be provided.
(7) Line 110: include “Arabidopsis thaliana” before cpi1-1 mutants (and delete “the”).
8) Line 112: delete “, and we detected…..plants”
(9) Lines 112-114: merge the sentences “The 
root length in cpi1-1 mutants expressing Flot1 was shorter than in the control seedlings (Figure 1B); it was 84% decreased compared with control seedlings (Figure 1C).” to e.g. “The 
root length of cpi1-1 mutants overexpressing Flot1 was reduced in 84% when compared to the control seedlings (Figure 1B, C)”; pay attention that as it is written, it seems that the decrease in root length is due to the cpi1-1 mutation + the overexpression of flot1!
(10) Lines 114-117: what “further insight” does the FM4-64 staining adds to the previous results (roots length)? From figures 1B and C we already know that cpi1-1 mutants have a decreased growth; here the expression patterns should be presented.
(11) Lines127-129: merge the two sentences as suggested above.
(12) Discussion: three paragraphs for the discussion seems too short; also this chapter is too descriptive; I would recommend a more elaborate discussion. Possible questions to be addressed: (1) the integrated role of flot a cpi and if possible a model with the protein networks involved in different biological processes; (2) what is the novelty of this work? (3) be more detailed and clear on how this work allows to draw conclusions on disease resistance (since the experiments did not include pathogenic or endosymbiotic infections, I have my doubts regarding the end of the discussion and the conclusion); (4) comparisons with studies in other plants are not clear.
(13) Experimental design: controls are missing, as you should have non-trasngenic (control) and transgenic (Flot1_GFP), for both wild-type and cpi1-1 mutants; otherwise, the data are not comparable. Please include the entire set of data.
(14) An extensive review of newly published paper should be included; although this topic is not much developed, there are more recent references than the ones cited: e.g. doi: 10.3389/fpls.2018.00991; 10.1093/pcp/pcz058; 10.1073/pnas; 10.1080/15592324.2017
Author Response
Response to reviewer 1
Manuscript ID: ijms-716938
Title: Dynamics and endocytosis of Flot1 in Arabidopsis require CPI1 function
Authors: Yangyang Cao, Qizouhong He, Zengxing Qi,Yan Zhang,Liang Lu,Jingyuan Xue,Junling Li,Ruili Li
Dear Reviewer,
We thank you for your highly positive comments on our manuscript. After reading these comments, we realized that there were some problems in our original manuscript and we found these comments and suggestions are extremely valuable for our revision.
As you will see, the issues raised have all been taken into account in the revised manuscript, which we explain in detail below.
Question 1: Gene names should be described in whenever a gene is cited for the first time, throughout the MS, e.g.: Line 15: - “Flot1” describe as you did for cpi1-1 Line 23: “flg22” – describe as for cip1-1
Reply: Yes, we fully agree with you. Therefore, as suggested, we have described the gene name of Flot1 and flg22 in the revised manuscript.
Question 2: Check the scientific names, e.g. In line 15 “Arabidopsis thaliana” is not in italic
Reply: We felt deeply sorry for the careless mistake. Therefore, we changed ‘Arabidopsis thaliana’ to ‘Arabidopsis thaliana’ in the revised manuscript.
Question 3: Line 37: “biological process”: use plural
Reply: We apologize that the writing is not appropriate. We have changed the “biological process” to “biological processes” in the revised manuscript.
Question 4: Lines 60-81: I would reduce this paragraph (is the detailed description of the role of sterol inhibitors relevant?) and transfer it to the beginning of the introduction, when the membrane microdomains are introduced (lines 32-34), before presenting the SPFH proteins. I think it makes more sense to first explain what are microdomains and their constitution and only then refer the protein networks involved in the different biological processes and the research gaps.
Reply: This is indeed an expert suggestion. However, the sterol inhibitors such as mβCD, Fen and PPMP deplete the sterols in the plasma membrane, resulting in the dynamics and endocytosis of the membrane proteins were affected. These results indicate that sterols in the membrane microdomains are important for the function of proteins. Therefore, this paragraph is demonstrated to be necessary for our hypothesis of the research. Furthermore, the specific effect of sterol-deficient mutants on Flot1 remains unclear and needs further study. So this paragraph focus on how sterols affect the function of some proteins which are localized in plasma membrane.
Question 5: The link between cpi1-1 and smt1 (lines 89-97) is not clear; please refer this aspect or delete the information on smt1; it would be more important to explain the link between flot and cpi proteins (and thus the hypothesis of your research).
Reply: We fully agreed with your comments that this aspect should be revised. As suggested, we have deleted the description of smt1 in this paragraph. Now this aspect focuses on the introduction of cpi1-1 mutants and the hypothesis between Flot1 and CPI in the revised manuscript.
Question 6: Lines 98-107: rather than summarizing the results, the rationale of the research should be provided.
Reply: We thank the reviewer for the valuable suggestion. Therefore, we have weakened the description of the experimental results and focus on introducing the rationale of experiment in the revised manuscript.
Question 7: Line 110: include “Arabidopsis thaliana” before cpi1-1 mutants (and delete “the”).
Reply: We thank the reviewer for this expert suggestion. As suggested, we added the “Arabidopsis thaliana” and deleted “the” in the revised manuscript.
Question 8: Line 112: delete “, and we detected...plants”
Reply: As suggested, we delete “, and we detected...plant” in the revised manuscript.
Question 9: Lines 112-114: merge the sentences “The root length in cpi1-1 mutants expressing Flot1 was shorter than in the control seedlings (Figure 1B); it was 84% decreased compared with control seedlings (Figure 1C).” to e.g. “The root length of cpi1-1 mutants overexpressing Flot1 was reduced in 84% when compared to the control seedlings (Figure 1B, C)”; pay attention that as it is written, it seems that the decrease in root length is due to the cpi1-1 mutation + the overexpression of flot1!
Reply: Thank you for your expert suggestions. Therefore, to make it easier to understand, we combine the two sentences into one as you suggested in the revised manuscript.
Question 10: Lines 114-117: what “further insight” does the FM4-64 staining adds to the previous results (roots length)? From figures 1B and C we already know that cpi1-1 mutants have a decreased growth; here the expression patterns should be presented.
Reply: Thanks for your comments. In order to further observe the root length of the plants at the cellular level, we stained the root cell membranes of the plants with FM4-64, this result further confirms the results at the cellular level. In addition, the expression patterns of Flot1 in cpi1-1 mutants are shown in the following Figure 2.
Question 11: Lines127-129: merge the two sentences as suggested above.
Reply: We agreed with your comment. Therefore, We have replaced the sentences “The number of cotyledon epidermal cells in control seedlings was also significantly lower than in cpi1-1 mutants; the number of cotyledon epidermal cells in cpi1-1 mutants was 153% of that in control seedlings (Figure 2C).” to “The number of cotyledon epidermal cells in cpi1-1 mutants was 153% of that in control seedlings (Figure 2C).”in the revised manuscript.
Question 12: Discussion: three paragraphs for the discussion seems too short; also this chapter is too descriptive; I would recommend a more elaborate discussion. Possible questions to be addressed: (1) the integrated role of flot a cpi and if possible a model with the protein networks involved in different biological processes; (2) what is the novelty of this work? (3) be more detailed and clear on how this work allows to draw conclusions on disease resistance (since the experiments did not include pathogenic or endosymbiotic infections, I have my doubts regarding the end of the discussion and the conclusion); (4) comparisons with studies in other plants are not clear.
Reply: Thank you for your expert comments. As you suggested, we expand and elaborate the discussion in the revised manuscript.
(1) In our study, we speculated that CPI was involved in Flot1-mediated defense response by inhibiting the endocytosis and coordinating dynamics of Flot1 proteins. However, according to our current experimental evidence, we still cannot speculate the specific relationship and regulation mechanism between the two proteins.
(2) The novelty of this work was based on our expertise in the field of single molecule analysis of plasma membrane protein dynamics together with our novel findings about the key role of sterol in plants. And we have added the advantages of these technologies in the discussion in the revised manuscript (Ehrhardt and Frommer, 2012; Miller et al., 2006).
(3) We apologize not to explain it clearly. In our experiment, we used flagelin22 (flg22) as pathogen infection factor to study whether CPI1-1 affect the function of Flot1. In fact, the immune response to flg22 of Flot1 in plants has been reported in our previous study (Yu et al., 2017). To make it clearer, we further describe the mechanism of disease resistance induced by flg22 in the discussion in the revised manuscript (Felix et al., 1999; Gravino et al., 2015; Navarro et al., 2006; Sun et al., 2013; Zipfel et al., 2004).
(4) Thanks for your helpful suggestion. Therefore, we added some research on other sterol mutants and compared them with our results in the revised manuscript.
Question 13: Experimental design: controls are missing, as you should have non-transgenic (control) and transgenic (Flot1_GFP), for both wild-type and cpi1-1 mutants; otherwise, the data are not comparable. Please include the entire set of data.
Reply: Thanks for your expert suggestions. It is well known that the control is vital for all the experiments. In fact, we measured the root length of non-transgenic plants in wild-type and cpi1-1 mutants. These results was similar to that was reported in the previous study (Men et al., 2008). Moreover, in the fluorescence observation experiment using variable-angle total internal reflection fluorescence microscopy (VA-TIRFM) and laser scanning confocal microscope (LSCM), we need to detect the fluorescence of Flot1-GFP. Therefore, we have the wild type expressing Flot1-GFP as the control group and cpi1-1 mutant expressing Flot1-GFP as the experimental group.
Question 14: An extensive review of newly published paper should be included; although this topic is not much developed, there are more recent references than the ones cited: e.g. doi: 10.3389/fpls.2018.00991; 10.1093/pcp/pcz058; 10.1073/pnas; 10.1080/15592324.2017
Reply: Thank you for your expert suggestions. We searched and read these newly references carefully, then we added them in the revised manuscript (Junkova et al., 2018; Shimada et al., 2019).
Overall, you will see that we have seriously revised the manuscript based on your suggestions. We are confident that our manuscript has been much improved. We sincerely hope that this revised manuscript can be finally accepted for publication.
Ehrhardt, D.W.; Frommer, W.B. New technologies for 21st century plant science. Plant Cell 2012, 24, 374-394.
Felix, G.; Duran, J.D.; Volko, S.; Boller, T. Plants have a sensitive perception system for the most conserved domain of bacterial flagellin. Plant J. 1999, 18, 265-276.
Gravino, M.; Savatin, D.V.; Macone, A.; De Lorenzo, G. Ethylene production in Botrytis cinerea- and oligogalacturonide-induced immunity requires calcium-dependent protein kinases. Plant J. 2015, 84, 1073-1086.
Junkova, P.; Danek, M.; Kocourkova, D.; Brouzdova, J.; Kroumanova, K.; Zelazny, E.; Janda, M.; Hynek, R.; Martinec, J.; Valentova, O. Mapping of plasma membrane proteins interacting with Arabidopsis thaliana Flotillin 2. Front. Plant Sci. 2018, 9, 991.
Men, S.; Boutte, Y.; Ikeda, Y.; Li, X.; Palme, K.; Stierhof, Y.D.; Hartmann, M.A.; Moritz, T.; Grebe, M. Sterol-dependent endocytosis mediates post-cytokinetic acquisition of PIN2 auxin efflux carrier polarity. Nat. Cell Biol. 2008, 10, 237-244.
Miller, A.E.; Fischer, A.J.; Laurence, T.; Hollars, C.W.; Saykally, R.J.; Lagarias, J.C.; Huser, T. Single-molecule dynamics of phytochrome-bound fluorophores probed by fluorescence correlation spectroscopy. Proc. Natl. Acad. Sci. USA 2006, 103, 11136-11141.
Navarro, L.; Dunoyer, P.; Jay, F.; Arnold, B.; Dharmasiri, N.; Estelle, M.; Voinnet, O.; Jones, J.D. A plant miRNA contributes to antibacterial resistance by repressing auxin signaling. Science 2006, 312, 436-439.
Shimada, T.L.; Betsuyaku, S.; Inada, N.; Ebine, K.; Fujimoto, M.; Uemura, T.; Takano, Y.; Fukuda, H.; Nakano, A.; Ueda, T. Enrichment of phosphatidylinositol 4,5-bisphosphate in the extra-invasive hyphal membrane promotes colletotrichum infection of Arabidopsis thaliana. Plant Cell Physiol. 2019, 60, 1514-1524.
Sun, Y.; Li, L.; Macho, A.P.; Han, Z.; Hu, Z.; Zipfel, C.; Zhou, J.M.; Chai, J. Structural basis for flg22-induced activation of the Arabidopsis FLS2-BAK1 immune complex. Science 2013, 342, 624-628.
Yu, M.; Liu, H.; Dong, Z.; Xiao, J.; Su, B.; Fan, L.; Komis, G.; Samaj, J.; Lin, J.; Li, R. The dynamics and endocytosis of Flot1 protein in response to flg22 in Arabidopsis. J. Plant Physiol. 2017, 215, 73-84.
Zipfel, C.; Robatzek, S.; Navarro, L.; Oakeley, E.J.; Jones, J.D.; Felix, G.; Boller, T. Bacterial disease resistance in Arabidopsis through flagellin perception. Nature 2004, 428, 764-767.
Reviewer 2 Report
The presented article is actual and informative, especially for the development of our understanding of plants resistance mechanisms. Nevertheless it requires some improvements.
Structure of the article is not traditional: the description of material and methods is placed in the end, which makes the understanding of the obtained results very difficult. Innovations are not always useful. In addition to that, it remained unknown how many seedlings were analyzed and how many tests were fulfilled. Also I would suggest adding the list of abbreviations used in the text.
